# Exploring the Effect of Balanced and Imbalanced Multi-Class Distribution Data and Sampling Techniques on Fruit-Tree Crop Classification Using Different Machine Learning Classifiers

Yingisani Chabalala [1,2,*] , Elhadi Adam [1] and Khalid Adem Ali [1,3]

1   Faculty of Science, School of Geography, Archaeology and Environmental Studies,
    University of Witwatersrand, Johannesburg 2000, South Africa
2   Department of Environmental Science, Science Campus, University of South Africa,
    Johannesburg 1710, South Africa
3   Department of Geology and Environmental Geosciences, College of Charleston, Charleston, SC 29424, USA
*   Correspondence: ywchabalala@gmail.com; Tel.: +27-725-899-389

**Abstract:** Fruit-tree crops generate food and income for local households and contribute to South Africa's gross domestic product. Timely and accurate phenotyping of fruit-tree crops is essential for innovating and achieving precision agriculture in the horticulture industry. Traditional methods for fruit-tree crop classification are time-consuming, costly, and often impossible to use for mapping heterogeneous horticulture systems. The application of remote sensing in smallholder agricultural landscapes is more promising. However, intercropping systems coupled with the presence of dispersed small agricultural fields that are characterized by common and uncommon crop types result in imbalanced samples, which may limit conventionally applied classification methods for phenotyping. This study assessed the influence of balanced and imbalanced multi-class distribution and data-sampling techniques on fruit-tree crop detection accuracy. Seven data samples were used as input to adaptive boosting (AdaBoost), gradient boosting (GB), random forest (RF), support vector machine (SVM), and eXtreme gradient boost (XGBoost) machine learning algorithms. A pixel-based approach was applied using Sentinel-2 (S2). The SVM algorithm produced the highest classification accuracy of 71%, compared with AdaBoost (67%), RF (65%), XGBoost (63%), and GB (62%), respectively. Individually, the majority of the crop types were classified with an F1 score of between 60% and 100%. In addition, the study assessed the effect of size and ratio of class imbalance in the training datasets on algorithms' sensitiveness and stability. The results show that the highest classification accuracy of 71% could be achieved from an imbalanced training dataset containing only 60% of the original dataset. The results also showed that S2 data could be successfully used to map fruit-tree crops and provide valuable information for subtropical crop management and precision agriculture in heterogeneous horticultural landscapes.

**Keywords:** horticulture; Sentinel-2 data; imbalanced data; sampling techniques; machine learning algorithms

## 1. Introduction

Agriculture is regarded as the foundation of all civilization, and crops are crucial to human nutrition and societal stability [1]. Due to the rising global human population, there is a need to closely monitor agricultural systems [2]. The human population of the world is experiencing a rapid increase—quadrupled in the last century, to *c.* 6.2 billion [3], and it is projected to be *c.* 8.7 billion by 2030 [4], and 10 billion by 2050 [5]. Therefore, food production needs to be doubled by 2050 [6]. Timely, efficient, and reliable agricultural data is of paramount importance to help monitor the demand for food, manage costs, and target agriculture-related policies [7,8]. Additionally, monitoring agricultural production is critical in terms of climate change, loss of biodiversity, and natural disasters—e.g., drought

and flood—which threaten food security at local and global scales [9]. The mapping of crops is key to the management of crops, estimation of yield, and food security [10,11].

Traditionally, information on the distribution of crops has been acquired through ground surveys and censuses [2]. However, field surveys are costly, outdated, cover small spatial extents, and are prone to human error [12]. Remote sensing (RS) technology, which has the advantages of monitoring extensive spatial extents and low cost, has substituted the traditional methods [12]. As RS data are advancing in terms of spatial and temporal resolutions, they are becoming increasingly important in generating crop-type maps [2]. Therefore, the need for timely, accurate, and reliable agricultural data has seen unprecedented growth in the economic importance of RS-based crop mapping over the last two decades [8]. The accessibility of satellite data by the public enabled many researchers to minimize ground surveys by creating low-cost crop-type maps—using features extracted from remotely sensed spectral differences in vegetation and other surfaces over time [13]. Agricultural RS offers insights that will radically change the operation and management of agricultural systems [14]. These efforts have been achieved through the successful application of machine learning (ML) algorithms, such as support vector machines (SVM), random forests (RF), and, increasingly, neural networks [15].

In addition to the number and quality of the training samples, algorithm performance may be affected by class imbalance [16,17]. Class imbalance occurs when the number of reference samples varies among the classes, leading to an imbalanced training set [16,17]. Most classifier learning algorithms that assume a fairly balanced distribution, such as $k$-nearest neighbor ($k$-NN), support vector machine (SVM), and random forest (RF), are affected by imbalanced training data [17]. Imbalanced data may occur in a deliberative sample, and they are also expected in random sampling [16,17]. In simple random sampling, the chance of choosing a class is related to the areal coverage of the class. Thus, relatively rare/minority classes will consist of smaller proportions of the training set, and vice versa. The original dataset was imbalanced despite efforts to add samples of fewer crop types—which compromises class-specific accuracy [18]. The imbalanced data problem worsens in a multi-class skewed distribution because the standard classifiers are designed for binary classification and assume a well-balanced class distribution [19]. Advanced boosting classifiers, such as gradient boosting (GB), adaptive boosting (AdaBoost), and eXtreme gradient boosting (XGBoost) are used to solve the limitations of standard classifiers.

Recently, the applicability of these boosting classifiers has been tested in crop type mapping [19,20] and land cover mapping [13]. For example, [21,22] tested the performance of the standard classifiers (RF, SVM) and boosting classifiers (GB, AdaBoost, XGBoost) in heterogeneous crop types mapping and reported a superior performance of 86.91% using XGBoost as compared to the RF and SVM. Similarly, [23] obtained the best accuracy metrics of 0.88% when forecasting cereal yield using the XGBoost classifier compared to SVM, RF, and multiple linear regression (MLR), whose accuracy was lower. The classification accuracies reported in these studies involved cereal, sugarcane, and wheat crops—they possess different canopy structures and spectral signatures. In this study, the crops were all fruit crops—namely, avocado, banana, guava, mango, and macadamia nut—which display almost similar spectral responses due to extreme heterogeneities. Such crop types are nearly impossible to discriminate using classical multi-spectral imagery analysis techniques [24]. This is made worse if the landscape is rugged terrains, intercropping systems scattered in small patches, and different crop area coverage/levels of growth with varying health statuses [25]. Additionally, the differences in management practices and soil fertility result in spectral similarities among crops, leading to uneven distribution of crop types [26].

Thus, different sampling techniques have been designed to solve the imbalanced class problem and ensure generalized data characteristics for class separability [27,28]. Such techniques include the modification of the classifier and data sampling [29]. The applicability of data sampling techniques, such as random oversampling (ROS) using the synthetic minority oversampling technique (SMOTE) and random undersampling (RUS), have been demonstrated in land cover mapping [28,29]. These techniques provide land cover, crop,

and soil mapping solutions by modifying the training dataset [30]. Although they produce acceptable classification accuracy, they fail to produce acceptable individual class accuracy [28,29]. Most of these techniques overfit the model or discard valuable samples required to determine the decision boundary between the classes [30–32]. In all cases/classification schemes, the datasets were split into test and training data before running the models. Running models before splitting datasets can allow identical samples to be present in both the test and training data, leading to the models overfitting the training data—the sampling techniques were only applied to training data to avoid model overfitting [18,19].

Accurate and timely mapping of crop types and having reliable information about the cultivation pattern/area play a key role in sustainable agriculture management. High-resolution, national-scale maps of agricultural land are needed to develop strategies for future sustainable agriculture. Thus, this study applied Sentinel-2 multispectral data and five widely-used ML classifiers—which were used for comparison—i.e., SVM, RF, an extreme gradient boosting machine called XGBoost, AdaBoost, and GBBoost, to map fruit crops and co-existing land use types in Levubu area, Limpopo province, South Africa, using imbalanced datasets.

The organization of the paper is as follows: Section 2 describes the materials and methods used in this research. Section 3 depicts the results of the paper, while the results of the research are discussed in Section 4. Finally, the conclusion of the paper is presented in Section 5.

## 2. Materials and Methods

### 2.1. Methods Overview

The potential of data sampling using Sentinel-2 multispectral data for crop type classification was studied by selecting the Levubu sub-tropical farming area as a research site. The Sentinel-2 image was pre-processed using SEN2COR in the ESA Snap platform to remove atmospheric effects, and five machine learning classifiers (i.e., GB, Ada Boost, RF, SVM, and XGB) were used to classify the processed image. Finally, the effectiveness of the classifier in classifying the monthly images was evaluated by computing different classification metrics, that is, the overall accuracy (OA), F1 score, user's accuracy (UA), and producer's accuracy (PA). A flowchart of the methodology followed in this research is presented in Figure 1.

### 2.2. Study Region

This research was conducted in Levubu, a subtropical fruit-tree crop farming region located at 23°4′60.00″ S and 30°16′60.00″ E in the Vhembe District, Limpopo, South Africa (Figure 2). The region experiences a warm subtropical climate with annual temperatures ranging from ~16 to 22 °C [33,34]. The annual rainfall ranges from ~200 to 1500 mm; however, it varies substantially owing to the geographic effect of the Soutpansberg mountain range [35,36]. Regions windward and east of the mountain range receive 200 to 400 mm, while regions leeward and west of the mountain range receive around 1000 mm [37]. The rainfall of the region is seasonal, with distinct wet and dry seasons ranging from October to March, and April to September, respectively [38].

Levubu, a smallholder agricultural region in Limpopo, South Africa, has diverse agro-climatic conditions that offer endless possibilities for commercial horticulture farming [39]. The dominant tree crop types are the macadamia nut, avocado (Hass, Maluma Hass, Pinkerton, and Fuerte), banana, guava, mango, and pine tree. In addition to the crop area, the region is characterized by other land-use types, such as water bodies, built-up areas, and woody vegetation. Most crops grown in the Levubu area are subtropical, and only a few citrus crops, such as oranges, are grown. Tree crops dominate the site, and small cultivated crops are grown depending on the season. Levubu crops are primarily rain-fed; however, during the dry season, crops are irrigated. Most of the tree crops bear fruit seasonally, and the flowering and harvesting time differs with each crop type. Many crops, including

the macadamia nut, banana, mango, and avocado, are exported to international markets (Department of Agriculture, Forestry and Fisheries) [40].

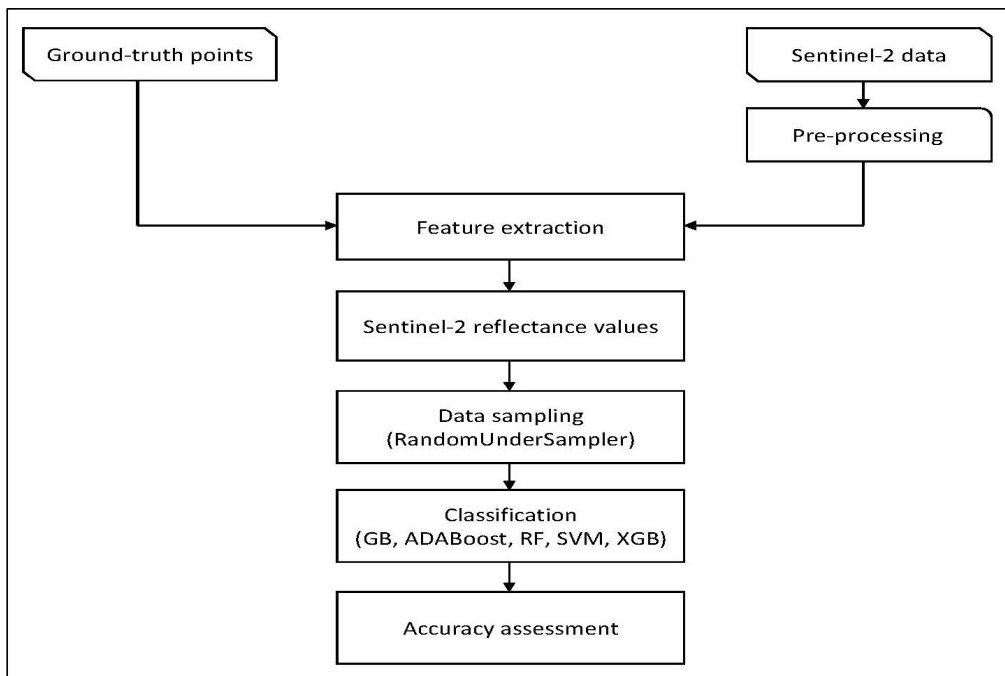

**Figure 1.** Flowchart of the methodology developed in this research.

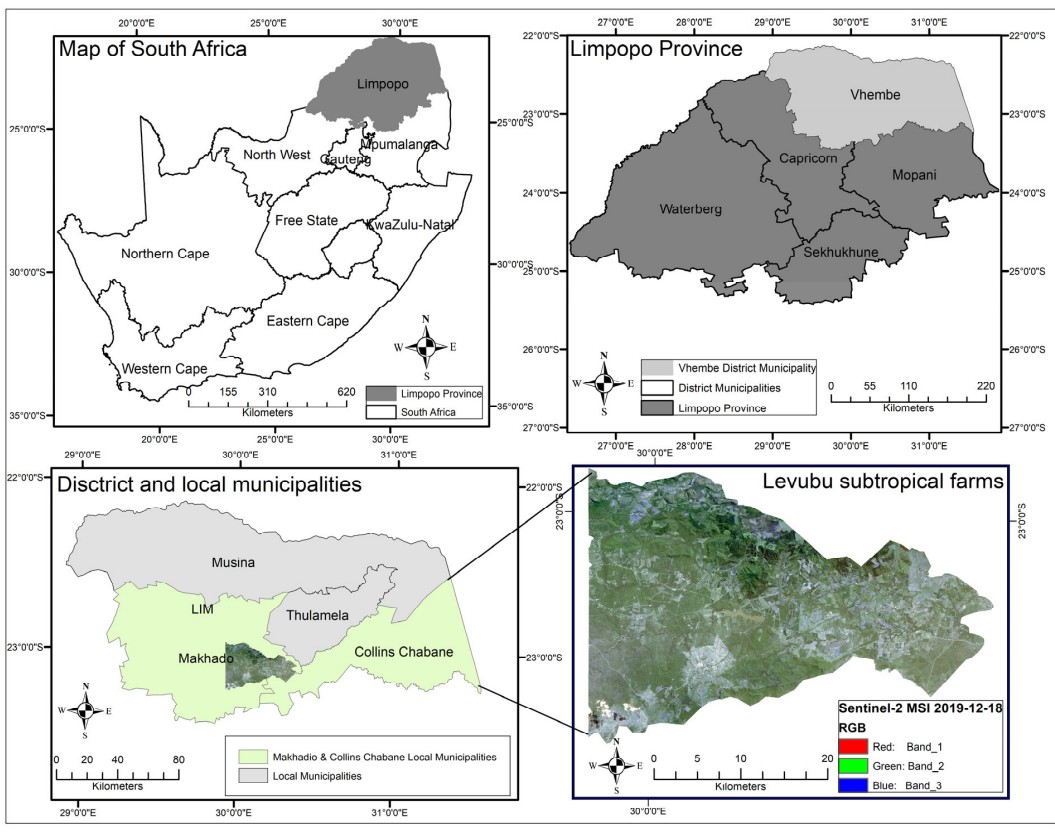

**Figure 2.** Map of the study region, depicting the Levubu subtropical farming region located in the Makhado and Collins Chabane Local Municipalities (lime green) of the Limpopo province (grey), South Africa.

### 2.3. Data Collection and Processing

### 2.3.1. Ground Truth Data

The ground truth data used for calibrating and validating fruit-tree crop maps and for applying ML algorithms in this study were collected using field surveys conducted on 28 and 29 December 2019 and 2 and 3 January 2020. A Garmin eTrex 20 Global Positioning System, with a positional accuracy of 5 m, was used to capture the geographic locations of dominant fruit-tree crops and co-existing land cover classes, namely: avocado, banana, bare soil, built-up, macadamia nut, mango, pine tree, water body, and woody vegetation. A combined total number of $n$ = 304 training samples was collected.

The collected field samples were used as a guide for visual comparison, identifying and labeling samples corresponding to the tree crop species and other co-existing land-use classes. A single-pixel sampling approach was applied when digitizing the training samples to select pure pixels and minimize the effect of landscape heterogeneity and mixed spectral signatures on classification accuracy [41]. A purposeful sampling method was used to increase the sample number and to consider the proportion of different types of fruit trees in the study area. The additional training samples were digitized using Google Earth Pro and ArcGIS version 10.6.1. The proportion of the farm was considered in generating the number and distribution of the samples, as depicted in Figure 2. The structure of the reference data is presented in Table 1, while the spatial distribution of the samples is depicted in Figure 3.

**Table 1.** Number of samples for fruit-tree crops and other existing land uses, the structure of training and testing data for each class used to map fruit-tree crops in Levubu subtropical farms.

| Tree Species and Other Land Use Classes | Reference Data | | Total |
|---|---|---|---|
| | Training | Testing | |
| Avocado | 109 | 47 | 156 |
| Banana | 181 | 77 | 258 |
| Bare soil | 120 | 52 | 172 |
| Built-up | 122 | 52 | 174 |
| Guava | 154 | 66 | 220 |
| Macadamia nut | 113 | 48 | 161 |
| Mango | 160 | 68 | 228 |
| Pine tree | 117 | 50 | 167 |
| Waterbody | 128 | 53 | 181 |
| Woody vegetation | 126 | 54 | 180 |
| Total sample size = 1897 | | | |

### 2.3.2. Data Sampling

Data sampling is crucial in supervised land cover mapping based on imbalanced data [42]. Previous research has dealt with class imbalance problems using data or algorithm-level methods [43]. The data-level methods modify the distribution and degree of imbalance of classes in the training dataset to fit the configurations of standard classifiers, while the algorithm-level methods modify the classifiers [43,44]. Two data modification methods used are undersampling and oversampling [32]. The former balances data by removing points from the majority class, while the latter duplicates points from the minority class [44,45]. Based on the existing remote sensing literature, much attention has been given to improving the accuracy of minority classes [29]. However, this study is built on the assumption that all crops are equally crucial in small-scale farming and must be recognized to allow efficient farm management practices [44]. Hence, the study employed undersampling techniques to retain the in-situ data and prevent the loss of crucial samples from the imbalanced dataset [46]. This was achieved by applying a RandomUnderSampler, a non-heuristic approach (naïve sampling) that selects samples from the majority class to balance the class distribution [44].

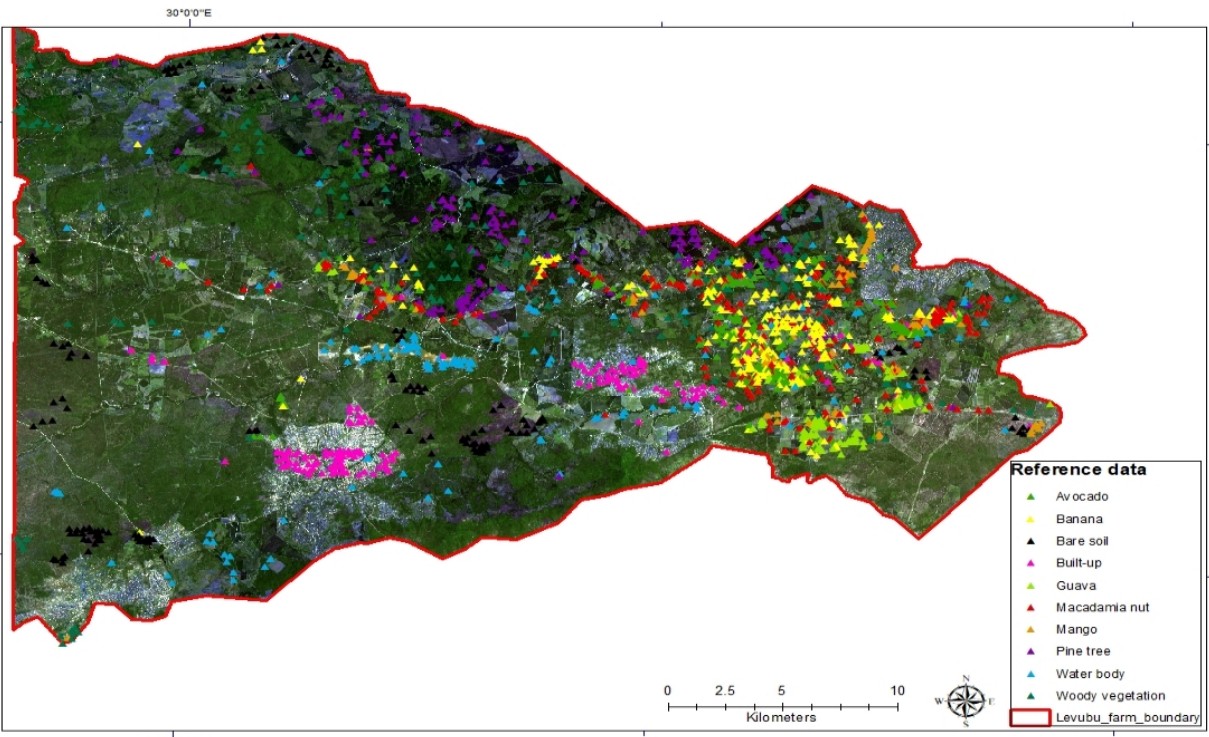

**Figure 3.** Location of fruit trees and the co-existing land-use types in the sub-tropical fruit farming area of Levubu. The background map represents a S2 RGB image obtained from European Space Agency.

In this study, the collected in-situ data is presented as Dataset 7 in Table 2 and consist of an imbalanced class ratio of 2:3, whereby the common (i.e., majority) species had more samples than the less common (i.e., minority) species. Six different sampling strategies were applied to generate new datasets from the original dataset, consisting of decreased imbalanced ratio and balanced class distribution (Table 1). Firstly, two balanced datasets (i.e., Dataset 1 and Dataset 2) were extracted, consisting of 100 and 150 balanced points.

Secondly, the imbalanced ratio on the original dataset was decreased by extracting four datasets using sampling strategies based on percentages (i.e., 40%, 50%, 60%, and 70%) and probability sampling. These strategies resulted in the formation of Dataset 3, Dataset 4, Dataset 5, and Dataset 6, respectively.

The sampling procedures were carried out using Imbalanced-Learn, a Python toolbox designed to handle problems associated with imbalanced datasets [47].

This toolbox consists of different libraries, where Scikit-Learn and RandomUnder-Sampler were used to sample the datasets in this study. Seven experiments of training datasets, including the original dataset, were tested to select the best model for mapping fruit-tree crops and co-existing land-use types in our study regions. The dataset was randomly divided into 70% training and 30% validation datasets for each class. All datasets were cross-validated to detect the behaviour of different classifiers on different samplings.

**Table 2.** List of fruit-tree crops and co-existing land-use types and the descriptive statistics of the undersampled and original datasets. The # stands for the number of the datasets used in this research.

| Crop Name | Abbreviation | Balanced Datasets (#1 and 2) | | Percentages Undersampled Datasets (#3–6) | | | | Unsampled Imbalanced Dataset (#7) |
|---|---|---|---|---|---|---|---|---|
| | | Dataset 1 | Dataset 2 | Dataset 3 | Dataset 4 | Dataset 5 | Dataset 6 | Dataset 7 |
| Avocado | AV | 100 | 150 | 62 | 78 | 94 | 109 | 156 |
| Banana | BN | 100 | 150 | 103 | 52 | 154 | 181 | 258 |
| Bare soil | BS | 100 | 150 | 69 | 86 | 103 | 120 | 172 |
| Guava | GV | 100 | 150 | 70 | 87 | 104 | 121 | 174 |
| Macadamia nut | MN | 100 | 150 | 88 | 110 | 132 | 154 | 220 |
| Mango | MG | 100 | 150 | 64 | 81 | 97 | 113 | 161 |
| Pine tree | PT | 100 | 150 | 91 | 114 | 137 | 160 | 228 |
| Built-up | BU | 100 | 150 | 69 | 86 | 100 | 117 | 167 |
| Waterbody | WB | 100 | 150 | 72 | 91 | 109 | 127 | 181 |
| Woody vegetation | WV | 100 | 150 | 72 | 90 | 108 | 126 | 180 |
| Total number of instances | | 1000 | 1500 | 760 | 875 | 1138 | 1328 | 1897 |

2.3.3. Remote Sensing Data Acquisition and Pre-Processing

Sentinel-2A (S2A) and Sentinel-2B (S2B) are a constellation of two polar-orbiting sensors placed in the same sun-synchronous orbit with the same sensor configurations. Both are equipped with a multi-spectral imaging (MSI) sensor. The imagery acquired from the two sensors is freely available from the European Space Agency (ESA) Copernicus Open Access Hub website (https://scihub.copernicus.eu/). The MSI sensor consists of 13 spectral bands ranging from visible and near-infrared (VNIR) to the shortwave infrared (SWIR) region (Table 2) [48]. The image has a wide-swath width of 290 km with 13 spectral bands consisting of three spatial resolutions (10 m, 20 m, 60 m). The 10 m spatial resolution bands (Blue: B2, Green: B3, Red: B4, NIR: B8) are centered at 490 nm, 560 nm, 665 nm, and 842 nm, and the 20 m spatial resolutions bands (Red Edge_1: B5, Red Edge_2: B6, Red Edge_3: B7, Narrow NIR: B8a, SWIR_1: B11, SWIR: B12) are located at 705 nm, 740 nm, 783 nm, 865 nm, 1,610 nm, and 2,190 nm. The 60 m spatial resolution bands (coastal aerosol: B1, water vapor: B9, SWIR-cirrus: B10) are located at 443 nm, 945 nm, and 1, 375 nm, respectively (Table 1). Additionally, it consists of two spectral bands that were systematically positioned in the red-edge position (REP). These bands are sensitive to chlorophyll content and are essential for crop mapping [35]. The temporal resolution between the two sensors is five days allowing for continuous crop mapping throughout the growing season [22].

Pre-processing of S2 data involves atmospheric corrections, geometric rectification, and radiometric calibration. In this study, the pre-processing was executed using a level 2A Prototype processor (SEN2COR) version 2.5.5 processing module in the ESA SNAP toolbox [35]. Specifically, the atmospheric/topographic correction for satellite imagery (AT-COR) algorithm within the SEN2COR toolbox was applied to correct the image [22,34]. All of the S2 band images were resampled to a 10 m spatial resolution using a nearest neighbor resampling method.

### 2.4. Machine Learning Classification Algorithms

Selecting an appropriate classifier for a classification problem remains challenging because the sensitivity of the classifiers during the learning stage varies per dataset owing to topography, land use, and class distribution [22]. The algorithms used in this study were selected based on their simplicity, robustness, and popularity in remote sensing of biophysical parameters. Therefore, it is imperative to elucidate their performance and consistency with previous research using the same acquisition condition but in a different environment. The study compares ML algorithms for mapping fruit-tree crops and co-existing land-use types in smallholder agriculture using balanced and imbalanced data. The classification was implemented with 70/30 split samples for training/testing. The analysis was carried out using Python Jupiter Notebook Version 3.9 (https://jupyter.org/).

2.4.1. Random Forest (RF)

The RF classifier is an ensemble tree-based classifier that randomly selects features from the training sample data and reduces tree correlations [49]. It uses bootstrap aggregating to build models using two parameters, ntree and mtry, interactively and independently by randomly selecting samples from the training dataset to make a final prediction [49]. All trees are aggregated to reach a final prediction. Thus, applying diverse decision trees grown using different random subsets reduces bias and prevents overfitting in the model [49]. The bootstrap aggregating (bagging) method is robust against model fitting and assists in obtaining a stable model [50].

A grid search was applied to select the optimal trees (ntree) and split nodes (mtry) used to map fruit-tree crops and co-existing land-use types from S2 data. Out of the calibration dataset, approximately 1/3 of the test data is held back and used to calculate unbiased error estimates, referred to as out-of-bag (OOB) [49]. The grid search approach was used for hyperparameter tuning, using all possible band combinations to yield a low OOB error [50]. Previous research has determined that the default values provide good

results [34,50]. Therefore, we utilized the default value of ntree = 500, and for mtry, the square root of the number of the variables as input into the RF model [49].

### 2.4.2. Support Vector Machine (SVM)

The SVMs are discriminative classifiers based on a statistical learning framework [51]. The SVM uses a margin-based classifier to identify optimal linear hyperplanes with high-significance class prediction using a kernel function [51].

The SVM consists of kernels used to model various classification problems as a hybrid classifier. Examples include the linear, polynomial, sigmoid, and radial basis function (RBF) kernels [52]. In this study, the SVM was applied using the function in Equation [1], which was implemented using the RBF kernel in Equation [3], consisting of two tuning parameters, termed the "gamma" (y) and "cost" (c) [22].

$$f(x) = \text{sign}\left(\sum_{i=1}^{n} \alpha_i y_i K(x, xi) + b\right), \tag{1}$$

where $\alpha_i$ illustrates the Lagrange multiplier and K $(x, x_i)$ represents the kernel fun.

### 2.4.3. Gradient Boosting (GB)

The GB classifier is an ensemble classifier that improves final prediction by combining residuals from prior weak models [53]. A gradient descent algorithm is used to sequentially train individual classifiers using equally weighted training data during the training process. The data are further re-calculated during the boosting process [53]. At the same time, the loss function is minimized by sequentially and iteratively adding new models to improve the base learners from the weak classifiers in an additive manner until the training data set is predicted perfectly [54].

### 2.4.4. Adaptive Boosting (AdaBoost)

AdaBoost is a meta-algorithm that can be used with other classifiers to improve classification accuracy [55]. It treats observations equally during optimization using stochastic GB machines [56]. The boosting process uses training data to build short decision trees in succession [57]. The performances of the trees are measured to determine the weight allocation during the next iteration [58].

The AdaBoost classifier is initiated by allocating weight on the training dataset based on how hard or easy the observations can be predicted. More weight is allocated to complex observations, while less is given to observations that can be easily predicted $(u_1, v_1)$, ... , $(u_n, v_n)$; $u_i \in U$, $v_i \in \{-1,+1\}$. At the start of the boosting process, all models are assigned the same distribution of $D_1(i) = 1/N$, i = 1, ... , N, where N represents the total number of observations [59]. The models are sequentially trained using the distribution $(D_t)$, and each model updates the weights of training instances of weak learners [60]. The new weights are calculated, and the distribution $D_t$: $D_{t+1}(i)$ is updated according to function 2. A final prediction is made based on the weighted vote accuracy of the trees on the training data [57].

$$\frac{D_{t(i) \exp(-a_t y_t T_t(u_t))}}{C_t} \tag{2}$$

### 2.4.5. eXtreme Gradient Boosting (XGBoost)

The XGBoost classifier has gained popularity in data science because it uses a gradient descent from the trees constructed in parallel to build an optimal model using the majority rule [54]. The XGBoost was developed to overcome limitations associated with previous boosting algorithms (i.e., GB and AdaBoost) [22]. It uses a regularized technique to control model overfitting while maximizing the model accuracy by sequentially adding models to correct errors from the existing models [21,61]. Furthermore, the classifier is an advanced gradient boosting that is computationally efficient and uses a cluster machine to continuously train large models while boosting fitted models on newly generated data [62].

The XGBoost contains different tuning hyper-parameters that are decided on using a grid search, making it robust and suitable for classifying imbalanced multi-class data from complex heterogeneous landscapes [21,61].

In this study, the XGBoost classification was performed using the XGBoost package with an objective function of "multi. softprob" to model the multi-class classification problem. The calibration of the models was carried out using stratified 10-fold cross-validation (CV). The CV is used to cater for the variance and enforce class distribution in imbalanced multi-class datasets while giving the dataset an equal chance to be returned to the testing set at fold iteration [63]. The learning rate was set to 0.01 with a base score of 0.5.

### 2.5. Mapping Accuracy Assessment

The prediction performance of the ML algorithms (GB, AdaBoost, SVM, RF, and XG-Boost) in mapping fruit-tree species and co-existing land use types using S2 was evaluated using 30% of the training samples from each of the seven models used in this study. A confusion matrix was used to calculate the threshold metrics widely used in imbalanced classification problems [64]. The OA, user's accuracy (UA), producer's accuracy (PA), and F-measures (F1-score) were used to evaluate the classifiers' performance [31,65]. The equations below were applied for the assessment.

$$OA = \frac{Sd}{n} \times 100\% \tag{3}$$

$$UA = \frac{X_{ij}}{X_j} \times 100\% \tag{4}$$

$$PA = \frac{X_{ij}}{X_i} \times 100\% \tag{5}$$

$$F_{score} = 2 \times \frac{UA \times PA}{UA + PA} \tag{6}$$

where $S_d$ demonstrates the samples correctly classified; n is the total number of validated samples; $X_{ij}$ represents the observation in row i of column j; $X_i$ is the marginal total of row i, and $X_j$ denotes the marginal of the total in column j.

The accuracy of the classifiers was then compared using a Student's paired t-test [66]. The paired t-test was applied with an alpha level of 5% ($\alpha = 0.05$), using the classifiers' OA to assess whether significant differences exist across the sampled datasets [67]. Boxplots were also created for visual interpretation to complement the t-test results.

## 3. Results

### 3.1. Spectral Separability of Fruit-Tree Crops and Co-Existing Land-Use Types

The mean spectral reflectance of the dominant fruit tree crops and co-existing land cover types identified during the field surveys are shown in Figure 4. Compared to the other nine classes, the built-up class has the highest reflectance values in the visible spectrum (Blue: 490 nm; Green: 560 nm; Red Edge_1:705 nm). The spectral reflectance values superficially on Red Edge_2, Red Edge_3, Red Edge_4, and NIR are higher for guava (GV), woody vegetation (WV), avocado (AV), macadamia nut (MN), mango (MG), banana (BN), and pine tree (PT) than for bare soil (BS) and water body (WB) (Figure 4). The GV has the highest spectral values in the REP among the fruit-tree crops, followed by the MN, MG, BN, and PT. For the non-crops, the built-up areas have the highest spectral values, followed by bare soil, and as expected, the waterbody has the lowest spectral values because of its high absorption.

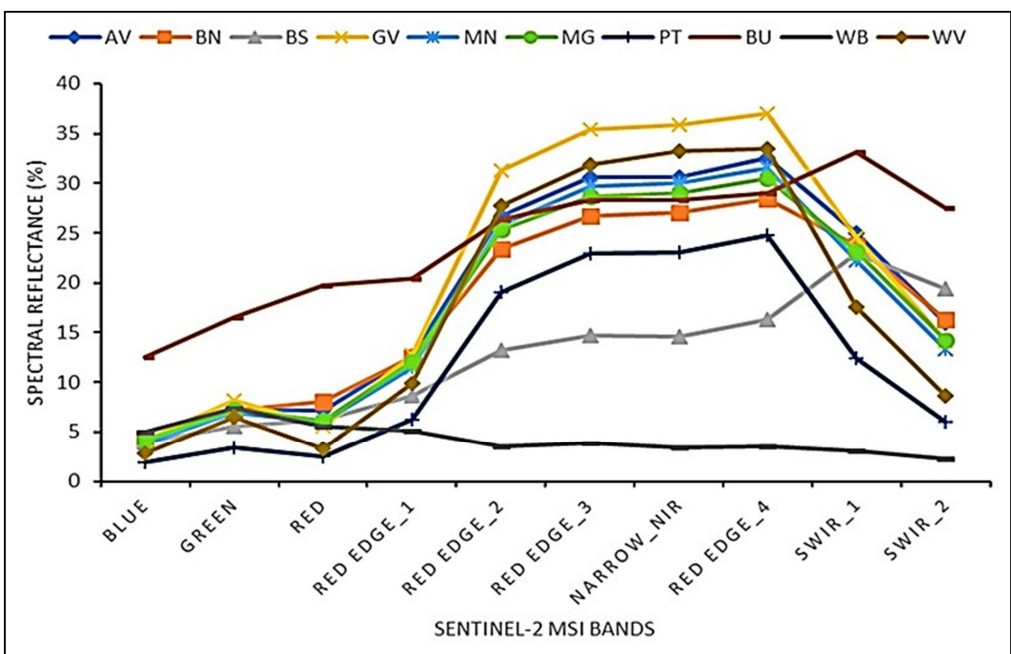

**Figure 4.** Spectral signature of fruit-tree crops, and co-existing land cover classes extracted from spectral Sentinel-2A algorithms image. The mapped classes are avocado (AV), banana (BN), bare soil (BS), guava (GV), macadamia nut (MN), mango (MG), pine tree (PT), built-up (BU), water body (WB), and woody vegetation (WV).

### 3.2. Fruit-Tree Crops Mapping Using Machine Learning Algorithms

This study used seven data sampling scenarios and single date S2 imagery for fruit-tree crop mapping by applying the various ML algorithms (RF, SVM, GBoost, AdaBoost, and XGBoost) (Figure 5a). The mapping results in Figure 5b show the effect of sampling techniques on the performance of the algorithms. The SVM classifier produced 71% and 69% OA at 150 balanced samples (Dataset_2) and 60% undersampling (Dataset_6). Next, the RF model recorded an OA of 65% at 100 balanced undersampled (Dataset_1) and 50% undersampling strategy (Dataset 4). The GB and XGBoost algorithms demonstrate the poorest performance for the data balanced at 100 sample points (Dataset_1) and sampled at 40% (Dataset_3) but recorded a slight improvement when each class has 150 balanced sample points (Dataset_2), scoring 50% and 70% for the undersampling strategies, respectively. Conversely, the performance of the AdaBoost classifier was stable across all datasets except at 40% undersampling (Dataset_3), in which the other classifiers also recorded accuracy scores of below 65%.

### 3.3. The Variable Importance

Variable selection is an important process that eliminates redundant information and increases the classifier's performance [42]. The permutation feature selection was used to retrieve important variable(s) from all possible parameter combinations [42]. The results show that the S2 bands' contribution and strength in discriminating classes vary across datasets, suggesting that the classifiers were sensitive to the sampling methods (Figure 6). For the 100 balanced samples, the SWIR_2 (B12: 2190 nm), SWIR_1 (B11: 1610 nm), and red (B4: 665 nm) are shown as the most optimal variables. For 150 balanced data points (Dataset_2), Red-Edge_2 (B6: 740 nm), green (B3: 560 nm), and SWIR_1 (B11: 1610 nm) were most important for discriminating the classes. A sampling at the 40% strategy, SWIR_2 (B12: 2190 nm), Red (B4: 665 nm), and RedEdge_2 (B6:740) produced the highest MDA scores, while at 50% sampling, the SWIR_1 (B11:1610), SWIR_2 (B12: 2190 nm) and Red-Edge_2 (B6: 740 nm) were the top contributing bands. At 60%, the SWIR_1 (B11: 1610 nm), SWIR_2 (B12: 2190 nm), and red (B4: 665 nm) were recorded as the best variables,

while, at 70% sampling, the Red-Edge_2 (B6: 740 nm), red (B4: 665 nm) and SWIR_2 (B12: 2,190 nm) were the best variables. The original dataset displayed the red (B4), SWIR_1 (B11: 1610 nm), SWIR_2 (B12: 2190 nm), and Red-Edge_2 (B6: 740 nm) to be the bands crucial for accurate mapping.

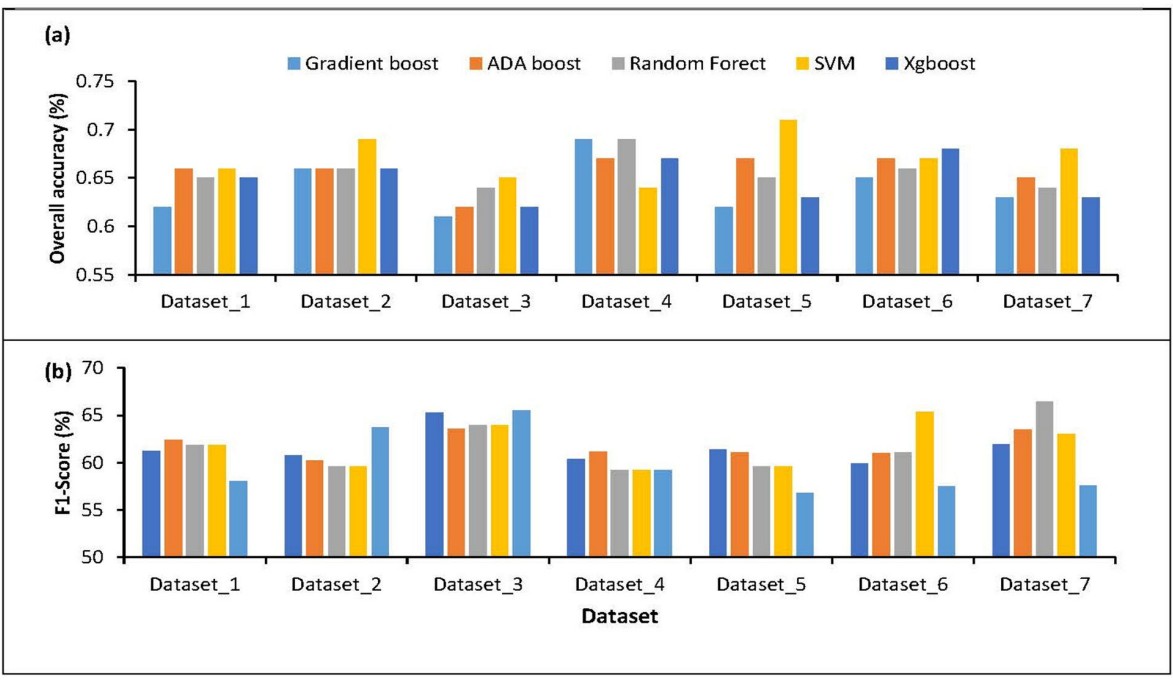

**Figure 5.** Comparison of the overall classification accuracies (**a**) and F1-Score (**b**) from GB, Ada Boost, RF, SVM, and XGBoost using seven different datasets. Dataset_1 (balanced undersampled at 100 points), Dataset_2 (balanced undersampled at 150 points), Dataset_3 (undersampled randomly at 40%), Dataset-4 (undersampled randomly at 50%), Dataset-5 (undersampled randomly at 60%), Dataset_6 (undersampled randomly at 70%), and Dataset-7 (imbalanced dataset).

### 3.4. The Class Accuracies

The F1 Scores

In Figure 7, the F1 scores for avocado, water body, and woody vegetation are above 60% for all ML classifiers across all datasets. Figure 8 shows that individual classes' UA and PA values were inconsistent across the datasets. The PA values for individual classes ranged between 17.78% (pine tree), using balanced data with 100 observations, to 100% (waterbody), using Dataset_2 (resampled at 40%) and the gradient boost classifier. At the same time, the UA ranged from 17.78% (pine tree, Dataset_2) to 100% for the water body class when applying the GB algorithm. The AdaBoost produced a PA of 23.33% for the mango class using Dataset_2 and 100% (waterbody) using Dataset_3. The UA accuracies ranged between 19.23% for macadamia nut, using Dataset_2, and 100% for waterbody, using Dataset_1 and Dataset_3 and the AdaBoost classifier.

The PA values for individual classes ranged from 0 (pine tree) to 100 (avocado, waterbody) for XGBoost. The UA accuracies ranged from 0% (pine tree) for Datasets 3, 5, and 6 to 100% for the avocado and waterbody classes for Dataset_3,5 and 6. The pine tree class recorded the lowest user accuracy on account of the low PA produced using the XGBoost classifier. The RF recorded PA values ranged between 15.38% for the mango class using Dataset_4 and 100% for the waterbody class using Dataset_5. The UA ranged between 32% for the pine tree class using Dataset_4 and 100% for the avocado and water body class using Dataset_1, 3, and 4, respectively. The SVM produced PA values ranging between 15.38% for the mango class using Dataset_4 and 100% for the waterbody class using Dataset_5.

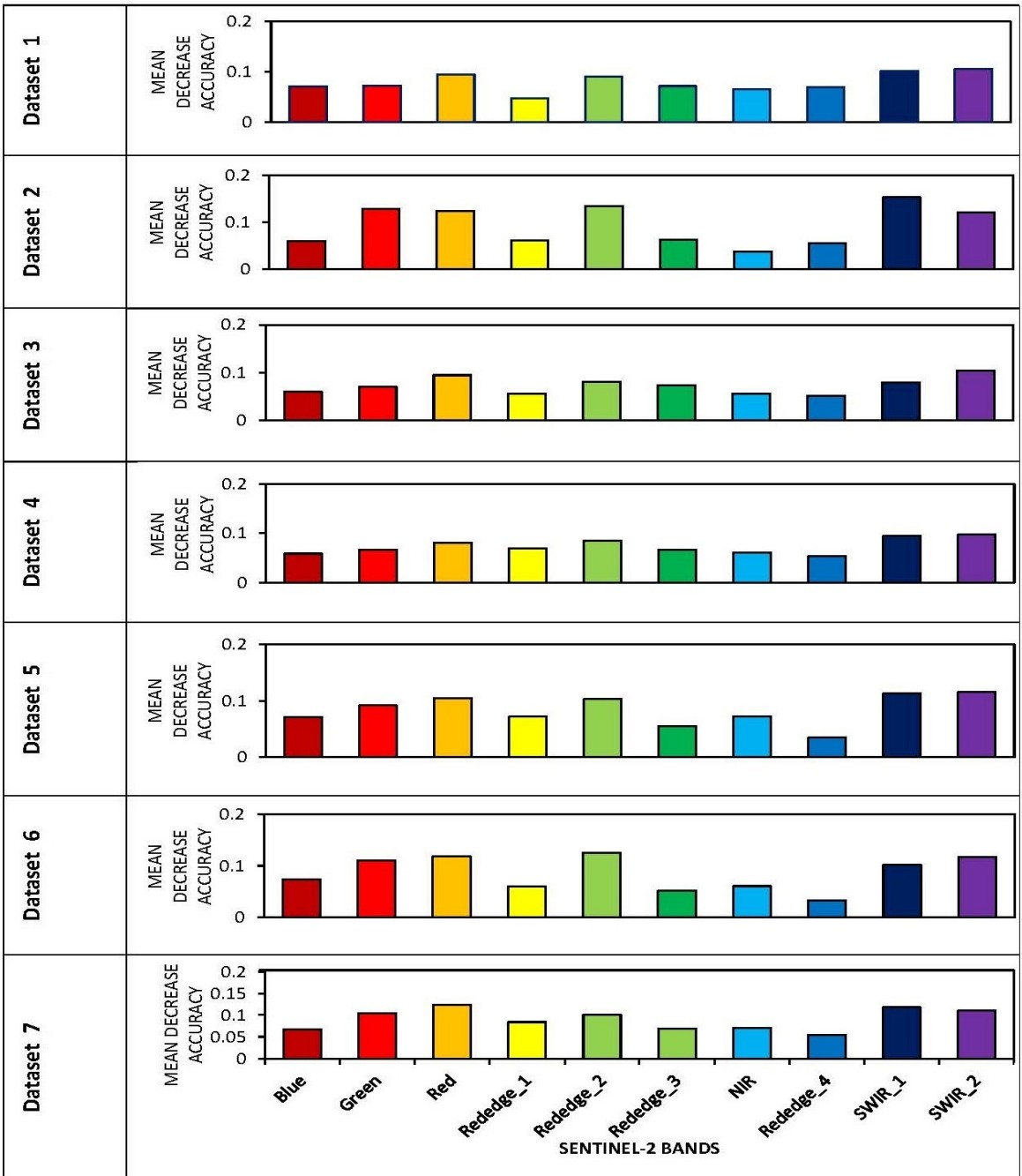

**Figure 6.** A comparative importance of explanatory variables used in permutation feature selection on seven datasets based on S2 spectral bands for Dataset 1 (balanced undersampled at 100 points), Dataset 2 (balanced undersampled at 150 points), Dataset 3 (undersampled randomly at 40%), Dataset 4 (undersampled randomly at 50%), Dataset 5 (undersampled randomly at 60%), Dataset 6 (undersampled randomly at 70%), and Dataset 7 (imbalanced dataset). The most influential bands are depicted by spikes indicating the highest feature importance scores. The color maroon, red, orange, yellow, lime green, green, light blue, blue, dark blue and purple represent Sentinel-2 bands, namely: Band 2 (Blue), Band 3 (Green), Band 4 (Red), Band 5 (Rededge_1), Band 6 (Rededge_2), Band 7 (Rededge_3), Band 8 (NIR), Band 8a (Rededge_4), Band 11 (SWIR_1) and Band 12 (SWIR_2).

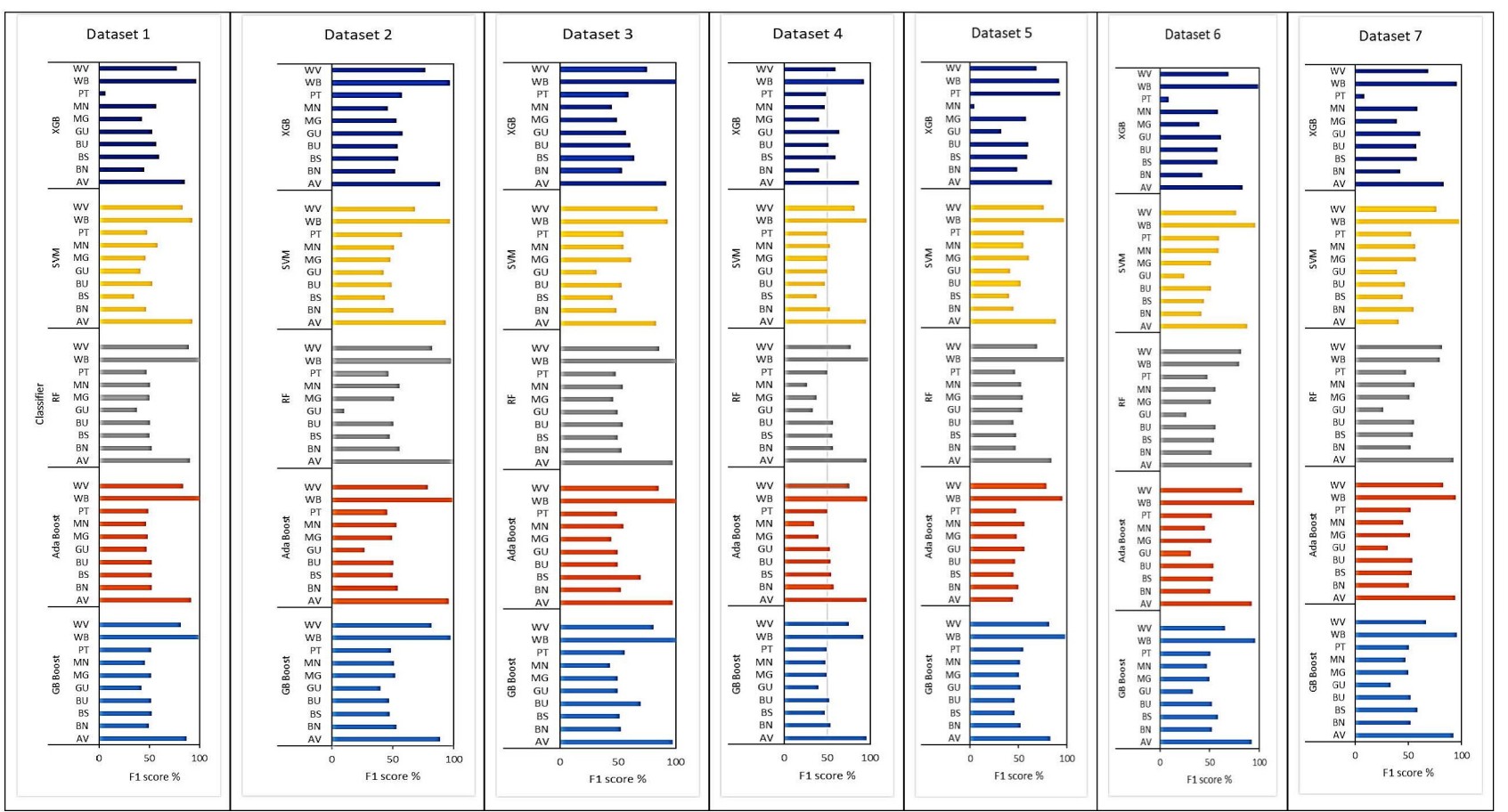

**Figure 7.** F1-scores rankings indicating the effect of sampling methods on class-specific accuracies obtained from GB, Ada Boost, RF, SVM, and XGBoost classifiers. The mapped classes are avocado (AV), banana (BN), bare soil (BS), built-up (BU), guava (GV), macadamia nut (MN), mango (MG), pine tree (PT), water body (WB), and woody vegetation (WV).

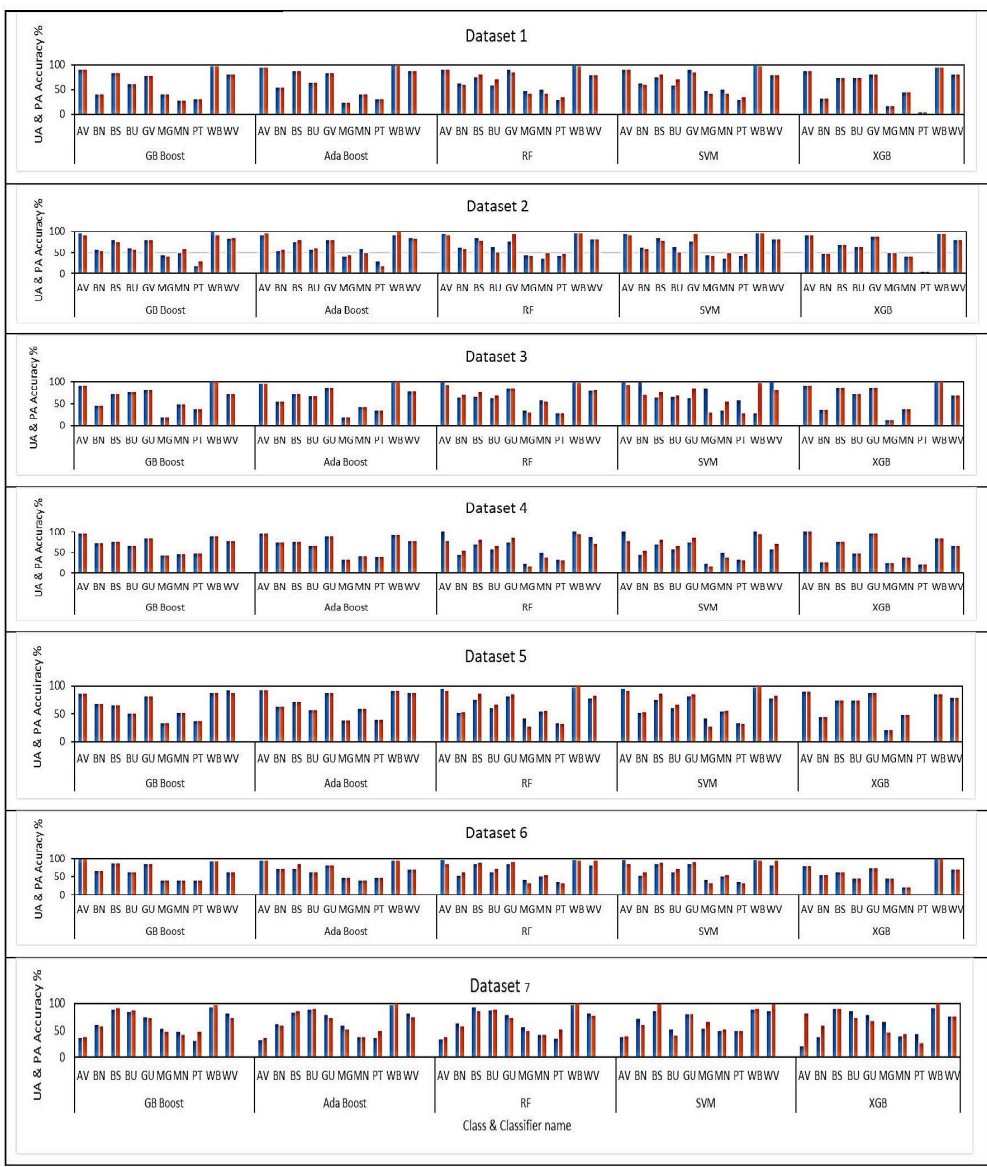

**Figure 8.** User's (blue) and producer's (red) accuracy per class produced using GB, Ada Boost, XG-Boost, RF, and the SVM classifier. On each classifier, there is a trade-off between the user and producer accuracy across the datasets.

### 3.5. Statistical Comparison of ML Classifiers from Seven Datasets

A comparison of the performance of the classifiers across the sampling techniques for the seven datasets shows that the Ada Boost and RF algorithms are sensitive (Figure 9). The overall mean ranged from 2.37 for RF and XGB ($p$ = 0.001) to 5.42 for SVM ($p$ = 0.008). The results indicate that the performance of the models across the datasets is statistically different and significant ($p$ = 0.001). However, the GB, SVM, and XGBoost algorithms have the same mean and $p$ of 0.292 and 0.780, respectively. The prediction of the fruit-tree crops with their co-existing land uses using different classifiers was significant across the datasets. Furthermore, the prediction was more statistically significant using the GB and XGBoost than for AdaBoost, RF, and SVM.

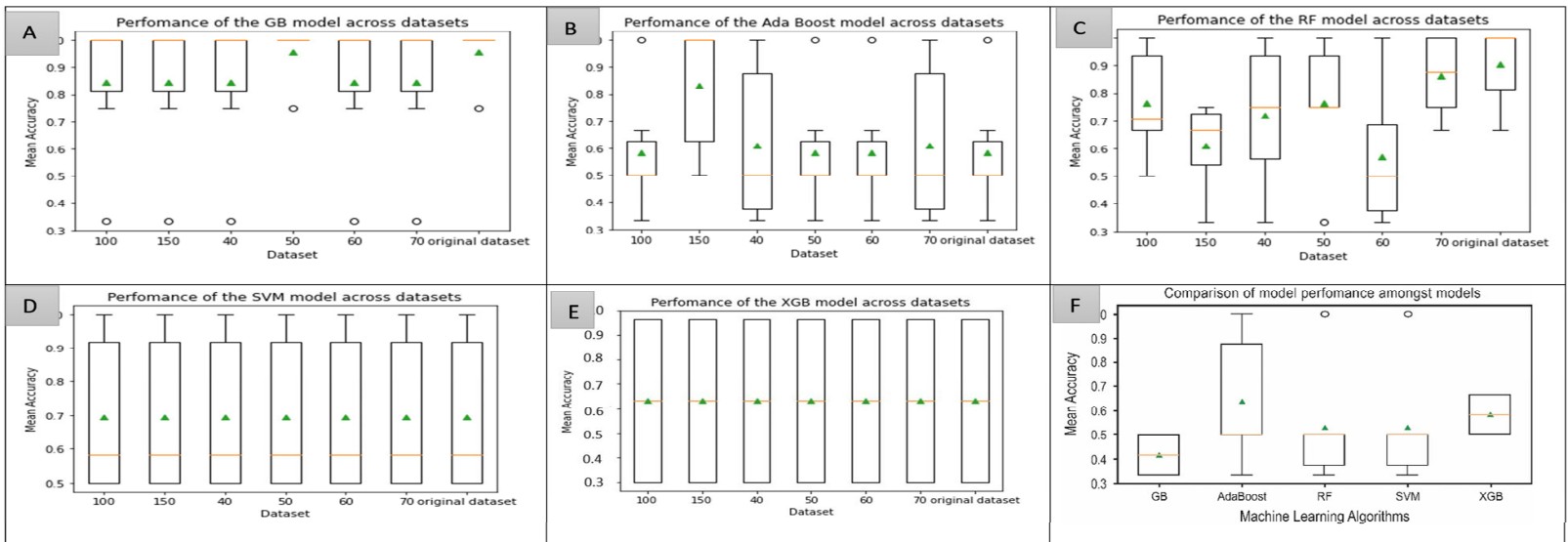

**Figure 9.** The boxplot shows the mean (yellow line), the low and upper quantile (black whiskers) comparison of the model performance across datasets and among the models using students paired *t*-test, where (**A**) represents the gradient boosting; (**B**) AdaBoost; (**C**) random forest; (**D**) support vector machines; (**E**) eXtreme gradient boosting; while (**F**) depicts the comparison of model performance amongst models, respectively. From the x-axis, the 100, 150, 40, 50, 60, 70, and original dataset labels correspond to Dataset 1, Dataset 2, Dataset 3, Dataset 4, Dataset 5, Dataset 6, and Dataset 7, respectively. The green triangles represent the mean in relation to the overall performances of each classifier per dataset.

## 4. Discussion

### 4.1. The Effect of Sampling Size on the Performance of Machine Learning Algorithms

Timely and accurate image classification from earth observation data involves ongoing efforts in various global agricultural systems based on distinctive climatic regions and different user needs, advances in mapping approaches, and easier access to remote sensing data. Spatially explicit information at a fine scale is required for site-specific management to optimize production and maximize returns. Thus, continuous mapping of smallholder agricultural systems is required for proper agricultural management, informing decisions, and developing policies to ensure sustainable food systems and economies in rural communities. This study evaluated the influence of sampling techniques on mapping fruit-tree crops in a subtropical agricultural region, namely, Levubu (South Africa), using a pixel-based approach with Sentinel-2 data.

The results demonstrate that applying effective sampling techniques and S2 data could provide spatially explicit information to improve the mapping of fruit-tree crops in a heterogeneous horticultural environment while reducing the need for extensive data collection. The fruit trees and co-existing land types in the study area were classified at an OA of 71% when applying the SVM algorithm, and the imbalanced dataset that was sampled at a 60% level (i.e., Dataset_6). Recent studies by [61,68] also reported the SVM as superior to other ML algorithms in discriminating crop species. The SVM uses spectrally weighted kernels to eliminate redundant information, select optimal variables based on their relative importance, and improve classification accuracy [41,61]. This advantage corroborates findings in previous crop type classification studies [63,69,70]. The ability of the SVM to achieve good classification results using small training samples should be considered when mapping landscapes with terrain characteristics that hinder the collection of balanced samples [71].

The superiority of the boosting classifiers (GB, AdaBoost, XGBoost) to the benchmark classifiers (RF, SVM) has been reported in previous studies [69,72]. The boosting algorithms have also been applied in crop mapping to overcome the configurations of the standard classifiers [21,22]. Studies by [23,73] also compared different ML algorithms for crop classification. They concluded that the XGBoost was the best classifier over SVM, RF, and GB. However, our results showed that the XGBoost algorithm performed poorly compared to SVM and RF for our study area. These results are consistent with findings from recent similar studies [21,61]. The performance of the GB classification showed higher accuracy when applied to a 50% sampled dataset (Dataset_4) but produced poor results for the other sampled datasets. This may be attributed to the sensitivity of the GB algorithm to hyperparameter selection, which was not evaluated in this study [61]. In all tested experiments, the results of the AdaBoost classifier were similar to the other classifiers, which contradicts with findings by [74]. In their study, the AdaBoost was more accurate in classifying the canola crops than the SVM.

The sensitivity of the classifiers differed based on the degree of class overlap, class imbalance, and lack of density in minority classes [75]. The effect of sampling on classifiers' performance was noticeable in imbalanced training samples. The GB, AdaBoost, and XGBoost were mostly affected compared to RF and SVM. The GB was affected by small balanced and imbalanced training samples (Dataset 1 and Dataset 3), while the overall accuracy of the AdaBoost, RF, and XGBoost decreased on small imbalanced training samples (i.e., Dataset 3). The performance of the XGBoost in this research was not stable on imbalanced samples, which contradicts the findings by [76]. The RF and SVM overall accuracy was over 65% on selected imbalanced training samples (i.e., Dataset 3, Dataset 5, Dataset 6, and Dataset 7). The performance of the RF was comparable on balanced samples. The differences in overall accuracy were significant with the increase in imbalanced training, which is similar to the report by [77]. The SVM was least sensitive to imbalanced training samples and performed well in detecting overlapping instances and showed the most accurate results even on small training samples. However, the results were slightly different from RF [16,77]. These findings are consistent with the results by [28]. We caution, how-

ever, that this conclusion is based on a small study area with a limited number of reference data. We suggest that the findings of this study should receive considerable additional investigation with other and big numbers of samples before it is accepted as a substitute for reliable results.

*4.2. The Statistical Comparison of the Classifiers among and within the Classifiers*

The five classifiers recorded significant differences ($p = 0.001$), suggesting that the sampling techniques had an effect on the classification results and can be used to improve crop discrimination in smallholder agriculture. In contrast, the mean differences in overall accuracies of the classifiers within the classifiers were statistically insignificant for GB, SVM, and XGBoost across the datasets ($p = 0.780$). Although the highest classification accuracy was obtained using the SVM model, the classifier's statistical comparison within the datasets was not statistically significant, corroborating the findings by [78,79], who stated that the SVM classifier was reliable and able to perform well in classification using limited training samples. The SVM classifier was robust and produced stable accuracies across the various balanced and unbalanced training set samples [68,77]. Conversely, the RF and Ada Boost performances differed across the various datasets and were statistically significant [80]. The models were robust and independent of the statistical distribution of training data, corroborating the statistical significance obtained using Student's paired *t*-test.

*4.3. Comparison of Individual Class Accuracies*

The discrimination between classes was assessed using the F1 scores, UA, and PA. As inferred from the F1 scores, it is evident that the performance of the classifiers is optimized when the data is balanced with a low imbalanced class ratio in the training data [63]. Although there was spectral overlapping among the mango, macadamia nut, and pine tree, the separability of the mango class from other classes was enhanced as a result of its particular structural and textural properties [81]. In addition, it has a loose canopy that displays unique tonal changes. However, these classes recorded low accuracies due to the differences in stem elongation, flowering, and fruiting, suggesting that the image acquisition was not optimal for these crops [82]. In some instances, spectral confusion can be exacerbated by factors, such as landscape heterogeneity, variations in agronomical practices, cropping calendars, and the pixel-based approach applied in this study [82]. Consistent with findings by [23,73], the best class-specific metrics in this study were achieved using GB, AdaBoost, RF, and XGBoost. The results indicate that the avocado class was the most accurately classified crop across the datasets. The co-existing land uses (i.e., water bodies and woody vegetation) have the highest classification accuracy except for the bare soil and pine tree classes. The performance of the classifiers was consistent and comparable in detecting the built-up class due to its high scattering mechanism, which was not affected by the balanced and imbalanced ratio of the datasets [83]. The lowest F1-score value was achieved using the SVM classifier, which is similar to the [58] findings. In their study, the highest F1 score was achieved using the RF algorithm when mapping the winter wheat crop in an urban agricultural region in Jiangsu Province, China. The crop spectral similarities had an effect on the classification algorithms as they lowered the classification results on some of the datasets [61]. High misclassifications among the macadamia nut, mango, and pine tree were observed from the accuracy metrics resulting from their small field size and spectral similarities. The difficulties in mapping macadamia trees are attributed to their hedgerow structures, as reported in other horticultural studies [82,83].

The pixel-based classification failed to capture the spectral heterogeneity in macadamia nuts and mango trees owing to intercropping, structural overlapping, and mixed spectral signatures [20,84]. The findings corroborate previous research that cited class overlapping and spectral similarities as the main factors hindering classification accuracy [84,85]. The results further demonstrate the effectiveness of S2 data in discriminating horticultural crops with mixed classes and high spatial heterogeneity [34,84]. Overall, stable class ac-

curacies were obtained using Datasets 3 and Dataset 4, indicating the robustness of the classifiers in handling imbalanced data.

### *4.4. Importance of Variables in Mapping Fruit-Tree Crops and Co-Existing Land-Use Types*

Measuring the importance of the explanatory variables concerning the modeled biophysical parameters has been fully explored in the existing literature [22,50]. In addition, variable importance is used to investigate the structure and the parameters that are used to train the model to understand their significance or a set of critical variables used to derive predictions [50]. The sensitivity of the S2 bands across different training datasets was evaluated using permutation feature selection. The comparative assessment of the derived feature importance (Figure 6) indicates a shifting pattern in the importance of S2 bands across different sampling techniques. The prominence of the S2 red band (B4: 665 nm), RedEdge_2 (B6: 740 nm), and the SWIR bands (B11,12: 1610–2190 nm) are comparable and more significant for class separability, regardless of the dataset used in this study. The results are consistent with the findings by [22]. The performance of RedEdge-1 (B5: 705 nm) is similar when data is balanced, sampled at 60%, 70%, and the original imbalanced dataset. The blue band (B2: 490 nm) scored high at 100 equal splits (balanced), sampling at 60% and 70%. The additional S2 RedEgde_2 band (B6: 740 nm) improved the classification results in the research area and was among the top three crucial variables. As expected and reported in previous studies, the results highlighted the effect of biophysical parameters on various electromagnetic spectra [50,86]. These findings are similar to previous research on crop type mapping using Sentinel-2 data [69,70]. The S2 shortwave region is characterized by increased spectral variability sensitive to water content on leaf canopies [72]. The impact of phenology on the SWIR reflectance enhanced the crop classification, as noted in recent research [87]. In contrast, the S2 visible spectrum and REP were found to be sensitive to chlorophyll content and leaf area index (LAI) in a study by [88]. The S2 10m spatial resolution reduces landscape heterogeneity and creates opportunities for crop type mapping in complex smallholder systems [89,90].

### 5. Conclusions

The influence of data sampling techniques and class imbalance for mapping fruit-tree crops and co-existing land uses was evaluated using five machine learning algorithms (GB, AdaBoost, RF, SVM, and XGBoost). The SVM algorithm outperformed the other algorithms with imbalanced data sampled at 60% and a balanced dataset with 150 instances per class. The overall accuracies ranged between 69% for 150 balanced datasets and 71% for imbalanced data sampled at a 60% level. Compared to GB, AdaBoost, RF, and XGBoost, the results show the superiority of the SVM classifier and Sentinel-2 data in mapping fruit-tree crops in a heterogeneous horticultural landscape.

It is concluded:

- Data sampling and selecting appropriate classification algorithms are essential for accurately mapping fruit trees in a horticultural environment characterized by complex and heterogeneous landscapes.
- Sentinel-2 offers similar classification accuracy and can be used for crop type inventories; these reduce the need for extensive data collection.
- The Sentinel-2 Red-Edge_2, SWIR_2, SWIR_1 (B11), and red (B4) bands are the most crucial predictor variables for crop classification using all datasets.
- The S2 Red-Edge bands are centered in the biomass region and contribute more to biomass studies.
- The best overall accuracy was achieved using the SVM and the dataset sampled at 60% (i.e., Dataset 7), while the class accuracies were stable when the dataset was sampled at 40% and 50%, respectively (i.e., Dataset 3 and Dataset 4).

**Author Contributions:** Data collection, conceptualization, methodology, analysis, report writing and editing, Y.C. Supervision, conceptualization, review and editing, E.A. Review and editing, K.A.A. The final version of the manuscripts has been approved by all authors. All authors have read and agreed to the published version of the manuscript.

**Funding:** This research forms part of Ph.D. research work funded by the University of Witwatersrand. No external funding was received from donors or funding institutions.

**Institutional Review Board Statement:** Not applicable.

**Informed Consent Statement:** Not applicable.

**Data Availability Statement:** The findings of this research are supported by data from European Space Agency which is openly available at https://scihub.copernicus.eu/.

**Acknowledgments:** Sincere thanks go to my supervisors, Elhadi Adam and Khalid Adem Ali, for their unwavering support and guidance. Your support and transfer of skills and knowledge will be forever cherished.

**Conflicts of Interest:** No conflict of interest were declared by the authors.

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
