# Peer review of "Exploring the Effect of Balanced and Imbalanced Multi-Class Distribution Data and Sampling Techniques on Fruit-Tree Crop Classification Using Different Machine Learning Classifiers"

_2673-7418, doi:10.3390/geomatics3010004_

Round 1

Reviewer 1 Report

This study assessed the influence of balanced and imbalanced multi-class distribution and sampling techniques on fruit-tree crops detection accuracy. The work is interesting, and the manuscript is well written and organized. I think there are still some issues to solve before the manuscript is accepted by Geomatics.

1. Different classifiers have different requirements on the number of samples. How does the number of samples affect classification accuracy?

 2. Spatial distribution of samples is also important for classification accuracy. What method is adopted in this study to arrange the spatial location of sample points? I think the figures of sample spatial distribution should be shown in the manuscript.

 3. Which classifiers are most affected by the imbalanced samples? Can you analyze it from the principle of classification algorithms?

4. What is the proportion of different types of fruit trees in the study area? Generally speaking, crops with small area need fewer sample points. Does the unbalanced sample selection in this study follow such a rule? When the samples are not balanced, whether the number of samples has a certain relationship with the proportions of fruit tree areas.

Author Response

Reviewer 1

1. Different classifiers have different requirements on the number of samples. How does the number of samples affect classification accuracy?

The reasons on how the number of samples affects classification accuracy are presented in section 4, sub-section 4.1, Line 506-520

 2. Spatial distribution of samples is also important for classification accuracy. What method is adopted in this study to arrange the spatial location of sample points? I think the figures of sample spatial distribution should be shown in the manuscript.

The spatial distribution of the sample is presented in Figure 1 section 2, subsection 2.2.1, Line 175, The samples were randomly selected from each tree species. We also used the purpose sampling method to increase the sample number and to consider the proportion of different types of fruit trees in the study area.

3. Which classifiers are most affected by the imbalanced samples? Can you analyze it from the principle of classification algorithms?

A paragraph was added in Section 4, subsection 4.1, Lines 506 to 520 with an analysis of how the classifiers were affected by imbalanced samples.

 What is the proportion of different types of fruit trees in the study area? Generally speaking, crops with small area need fewer sample points. Does the unbalanced sample selection in this study follow such a rule? When the samples are not balanced, whether the number of samples has a certain relationship with the proportions of fruit tree area

Yes, the proportion of the farm was considered in generating the number and distribution of samples. This has been explained in the ground truth data collection Section 2, Sub-Section 2.2.1, Lines 171 to 176

Reviewer 2 Report

The paper is technically sound and the experimental results are satisfactory. However, the paper needs improvement in terms of formatting, alignment, and spelling mistakes. My specific comments are as follows.

1. The title of the paper should have been smaller.

2. Line 64, space missing between [24] and 'obtained'.

3. In Line 95, there is an unnecessary ']' character exists.

4. What is the need for a ';' at the end of line 125 and in line 144?

5. Authors should add an extra paragraph at the end of the Introduction to briefly discuss the different sections of the paper.

6. In Figure 2, sub-figure labels are hard to see even after zooming in. Provide better-quality images.

7. Line 172, 'n=304'. What is n? Define it.

8. Alignment mismatch: Section 2.3.3, Equation [2]. Correct them.

9. Line 220, Multi-Spectral Imager or Multi-spectral Imaging?

10. Poor placement of Equations [3], [4], [5], and [6].

11. In Section 2, sub-section headings (titles) are bold but in Sections 3 and 4, sub-section headings are not bold, why?

12. Try to fit Figure 5 and its caption on a single page.

13. Poor alignment of Section 3.4 (and 3.4.1).

14. Revise the entire paper for rectifying misalignments, and spelling mistakes.

Author Response

Reviewer 2

1. The title of the paper should have been smaller.

The title has been shortened to read “Exploring the effect of balanced and imbalanced multi-class distribution on fruit-tree crop classification using different machine learning classifiers”

2. Line 64, space missing between [24] and 'obtained'.

The space between  [24] and obtained has been created.

3. In Line 95, there is an unnecessary ']' character exists.

The unnecessary character has been removed

4. What is the need for a ';' at the end of line 125 and in line 144?

The semi-colons ‘;’ i.n line 125 and line 144 have been removed.

5. Authors should add an extra paragraph at the end of the Introduction to briefly discuss the different sections of the paper.

An extra paragraph that describes the section of the paper has been added as suggested by the reviewer.

6. In Figure 2, sub-figure labels are hard to see even after zooming in. Provide better-quality images.

The labels of the maps in figure 2 have been increased for visibility. All figures have been saved as 600 dpi.

7. Line 172, 'n=304'. What is n? Define it.

The sentence has been revised and it now reads as “A combined total number of n= 304 training samples was collected”.

8. Alignment mismatch: Section 2.3.3, Equation [2]. Correct them.

The alignment of equation [2] has been fixed

9. Line 220, Multi-Spectral Imager or Multi-spectral Imaging?

The words Multi-Spectral Imager were changed to Multi-Spectral Imaging (Line 220)

10. Poor placement of Equations [3], [4], [5], and [6].

The placements of Equations 3 to 6 have been fixed.

11. In Section 2, sub-section headings (titles) are bold but in Sections 3 and 4, sub-section headings are not bold, why?

The boldness in section 2 sub-section headings has been removed for consistency.

12. Try to fit Figure 5 and its caption on a single page.

The format of Figure 5 was changed to fit its caption on a single page

13. Poor alignment of Section 3.4 (and 3.4.1).

The alignment in section 3.4.1 of 3.4 has been fixed.

14. Revise the entire paper for rectifying misalignments, and spelling mistakes.

The entire paper has been revised and misalignments and spelling mistakes have been corrected.

Reviewer 3 Report

1.      The article citation format is wrong. I think it's better to quote sequentially.

2.      The structure of introduction should be improved. It is difficult to get the aim of this research clear.

3.      Figures 1-2 are not clear. It is hard to recognize the fronts.

4.      The quality of Figures3-6 is low. Please improve them.

5.      The tables in the article should be adjusted, at least the font should be consistent.

6.      The formula format is confusing, please correct it.

7.      RF, SVM, GB, AdaBoost and XGBoost are widely used machine learning algorithms. It is not necessary to explain them in detail.

8.      In the section of Results, in addition to giving experimental results, results analysis should also be given. For example, why do built-up areas have the highest spectral values? It is better to explain the reasons. In section 3.2, form the results, which ML algorithm is the most suitable for fruit-tree crops mapping? Which dataset holds the highest accuracy?

9.      ML requires a lot of samples. The total number of samples is about 1000-2000, is the results of small sample data reliable?

10.   The information in Fig.6 and Fig.7 is too much.

11.   Why you use Sentinel-2 datasets ,rather than landsat-8? Please explain it.

12.   I think there are too many references for a journal article

Author Response

Reviewer 3

1.      The article citation format is wrong. I think it's better to quote sequentially.

The sequence of citations has been fixed.

2.      The structure of introduction should be improved. It is difficult to get the aim of this research clear.

The structure of the introduction was improved to allow readers to easily identify the aim of the research.

3.      Figures 1-2 are not clear. It is hard to recognize the fronts.

The fonts in Figures 1 and 2 have been fixed to increase readability.

4.      The quality of Figures 3-6 is low. Please improve them.

The quality of Figure 3-6 has been increased to allow easier interpretations.

5.      The tables in the article should be adjusted, at least the font should be consistent.

The tables have been adjusted and the same font and size were used for consistency.

6.      The formula format is confusing, please correct it.

The format of all formulas has been corrected

7.      RF, SVM, GB, AdaBoost and XGBoost are widely used machine learning algorithms. It is not necessary to explain them in detail.

We agree with the reviewer; however, we decided to include this information to provide an overview of the classifiers to enable readers outside of this field so they can understand how the methods work.

8.      In the section of Results, in addition to giving experimental results, results analysis should also be given. For example, why do built-up areas have the highest spectral values? It is better to explain the reasons. In section 3.2, form the results, which ML algorithm is the most suitable for fruit-tree crops mapping? Which dataset holds the highest accuracy?

We have provided results analysis indicating reasons why certain classes like built-up have the highest spectral values (Section 4, sub-section 4.1, Line 506-522). Furthermore, information on which ML classifier was suitable for fruit-tree crop mapping is provided in Section 4, Sub-section 4.1, Lines 483-484. We also indicated which dataset holds the highest accuracy (Section 4, Line 483-484).

9.      ML requires a lot of samples. The total number of samples is about 1000-2000, is the results of small sample data reliable?

We agree with the reviewer. The study area is relatively small with limited access. However, we believe that the number of samples used here is adequate to test the effect of balanced and imbalanced data in a heterogenous agricultural system. We have noted the limitation of the sample number in the discussion section, Section 4, Sub-Sevtion 4.1, Lines 522-526.

10.   The information in Fig.6 and Fig.7 is too much.

We agree with the reviewer. However, due to the amount of work done in this study, there is no other best way to present the results than the way it is. The study compared five machine learning algorithms using seven datasets.

11.   Why you use Sentinel-2 datasets ,rather than landsat-8? Please explain it.

Sentinel-2 has finer spatial and temporal resolution than Landsat. Sentinel-2 has more spatial resolution providing additional spatial information and red-edge bands which are sensitive to chlorophyll content and have been proven to increase the detection accuracy of fruit-tree species than Landsat. Sentinel-2 has visible and Near Infrared bands that are narrower and broader, respectively. Studies that conducted a comparative analysis of the performance of Landsat and Sentinel-2, reported superior results from Sentinel-2 than with Landsat (Clark. 2017; Wang et al., 2022). This study tested the applicability of Sentinel-2 variables in classifying fruit trees and co-existing land use types in a heterogeneous smallholder region with different farming and management practices.

12.   I think there are too many references for a journal article

The in-text citations have been reduced by cutting off the redundant citations to a minimum.

Round 2

Reviewer 3 Report

(1)Formula number is not uniform. Please carefully check and correct. 

(2) The front in figures is not the same as the text front. Please correct.